# Trimetallic Chalcogenide Species: Synthesis, Structures, and Bonding

**DOI:** 10.3390/molecules27217473

**Published:** 2022-11-02

**Authors:** Sourav Kar, Debipada Chatterjee, Jean-François Halet, Sundargopal Ghosh

**Affiliations:** 1Department of Chemistry, Indian Institute of Technology Madras, Chennai 600036, India; 2Laboratory for Innovative Key Materials and Structures (LINK), IRL 3629, CNRS-Saint-Gobain-NIMS, National Institute for Materials Science (NIMS), Tsukuba 305-0044, Japan

**Keywords:** aromaticity, boron, cubane, selenium, sulfur

## Abstract

In an attempt to isolate boron-containing tri-niobium polychalcogenide species, we have carried out prolonged thermolysis reactions of [Cp*NbCl_4_] (Cp* = *ɳ*^5^-C_5_Me_5_) with four equivalents of Li[BH_2_E_3_] (E = Se or S). In the case of the heavier chalcogen (Se), the reaction led to the isolation of the tri-niobium cubane-like cluster [(NbCp*)_3_(*μ*_3_-Se)_3_(BH)(*μ*-Se)_3_] (**1**) and the homocubane-like cluster [(NbCp*)_3_(*μ*_3_-Se)_3_(*μ*-Se)_3_(BH)(*μ*-Se)] (**2**). Interestingly, the tri-niobium framework of **1** stabilizes a selenaborate {Se_3_BH}^−^ ligand. A selenium atom is further introduced between boron and one of the selenium atoms of **1** to yield cluster **2**. On the other hand, the reaction with the sulfur-containing borate adduct [LiBH_2_S_3_] afforded the trimetallic clusters [(NbCp*)_3_(*μ*-S)_4_{*μ*-S_2_(BH)}] (**3**) and [(NbCp*)_3_(*μ*-S)_4_{*μ*-S_2_(S)}] (**4**). Both clusters **3** and **4** have an Nb_3_S_6_ core, which further stabilizes {BH} and mono-sulfur units, respectively, through bi-chalcogen coordination. All of these species were characterized by ^11^B{^1^H}, ^1^H, and ^13^C{^1^H} NMR spectroscopy, mass spectrometry, infrared (IR) spectroscopy, and single-crystal X-ray crystallography. Moreover, theoretical investigations revealed that the triangular Nb_3_ framework is aromatic in nature and plays a vital role in the stabilization of the borate, borane, and chalcogen units.

## 1. Introduction

Metal-metal bonding is a constantly growing and widely studied area of chemistry [1,2,3,4]. The presence of d-orbitals in transition metals endows a much more diverse structure, chemistry, and chemical bonding than non-metal elements [5]. Paired metals can be linked by single [1,6,7,8], double [1,9], triple [9,10], quadruple [11], and in a few cases, quintuple bonds [12,13]. These types of metal–metal bonds are formed through σ, π, and δ-bonding interactions. With the exception of multiple bonding, metal atoms can form different types of frameworks, such as triangular, square, and three-dimensional clusters [14,15,16,17,18,19,20]. These metal frameworks ([Fig molecules-27-07473-ch001]) can show various types of aromaticity, such as σ-, π-, and δ-type. Recent findings of aromaticity in metal clusters have advanced the description of the electronic structures, chemical bonding, and stability of transition-metal clusters [21,22,23,24,25,26,27]. However, aromatic metal clusters are often observed in the gas phase, with the exception of a few cases [28,29]. For example, Robinson et al. structurally characterized the first ever main-group metal 6-electron aromatic cluster (see A in [Fig molecules-27-07473-ch001]) with π-aromatic properties, in 1995 [30]. Later, Maestri et al. isolated the transition-metal containing 2-electron π-aromatic cluster, namely [{Pd(PPh_3_)(SPh)}_3_]^+^ (C, [Fig molecules-27-07473-ch001]) [31]. On the other hand, Sadighi et al. isolated a 2-electron tri-gold monocation (B, [Fig molecules-27-07473-ch001]) showing σ-aromaticity [32]. Freitag et al. enriched a series of 2-electron σ-aromatic metal clusters with the isolation of the triangular clusters [Zn_3_Cp*_3_]^+^ (D, [Fig molecules-27-07473-ch001]) and [Zn_2_CuCp*_3_] [33]. Recently, Liddle et al. synthesized a 2-electron tri-thorium cluster [{Th(*η*^8^-C_8_H_8_)(*μ*-Cl)_2_}_3_{K(THF)_2_}_2_]_∞_ (E, [Fig molecules-27-07473-ch001]) with σ-aromatic metal-metal bonding [34]. These types of unique bonding and aromaticity of metal frameworks may lead to the stabilization of many main group unstable species.

Many main group species that are quite unstable and not experimentally achieved as free species are isolated utilizing different types of metal frameworks [7,35,36,37,38]. For example, borylenes (B-R) can only be trapped by mono, bi, or tri- metallic frameworks [7,35]. In addition, many boranes and their analogues, from small boranes to higher nuclearity clusters, are stabilized in the coordination sphere of metals with structural and bonding diversity [7,14,36,39,40,41,42,43]. For example, we have recently isolated diborene(2) and diborane(4) utilizing group six trimetallic and bimetallic skeletons, respectively [44]. The first ever classical diborane(5) was also isolated utilizing a bimetallic coordination sphere of Ta atoms [45]. Further, a di-titanium framework allowed us to isolate the unique structural motif of hexa-borane [B_6_H_6_], i.e., a hexagonal flat borane ring [8]. With the exception of borane, many other main group species are isolated utilizing metal’s coordination sphere [46,47,48,49]. Further, we and others have isolated a series of group five trimetallic systems, which enabled us to isolate the thioborate ligand {BS_3_H}^−^ and its analogues [50,51,52,53,54]. In order to isolate small borane/borate and chalcogen species, we have extended our studies with niobium metal and heavier chalcogens. Herein, we present and discuss the isolation and structural explication of various trimetallic polychalcogenide frameworks, which stabilized tri/tetra-coordinated boron species and monochalcogen species. Theoretical calculations revealed the stability, bonding, and aromaticity of these trimetallic species.

## 2. Results and Discussion

### 2.1. Reactivity of [Cp*NbCl_4_] with Li[BH_2_Se_3_]

As shown in Figure 1, the prolonged thermolysis reaction of [Cp*NbCl_4_] with four equivalents of Li[BH_2_Se_3_] led to the formation of brown solid **1** and green solid **2**. The ^1^H NMR spectrum of **1** showed a single peak at *δ* = 2.21 ppm, which suggests the existence of a single Cp* environment. On the other hand, the ^1^H NMR spectrum of **2** showed two very close resonances that appeared at *δ* = 2.18 ppm in a 2:1 ratio, which suggests the existence of two different Cp* environments. The presence of one and two Cp* environments in **1** and **2**, respectively, was further confirmed by ^13^C{^1^H} NMR spectra. In addition, broad chemical shifts at *δ* = 3.87 and 3.65 ppm in ^1^H NMR of **1** and **2**, respectively, suggested the presence of terminal B-*H* protons in **1** and **2**. Moreover, the IR spectroscopy of **1** and **2** confirmed the existence of this terminal B–*H* proton. The ^11^B{^1^H} NMR spectrum of **1** showed a single resonance that appeared at *δ* = −8.3 ppm, whereas a chemical shift at *δ* = −6.4 ppm has been observed in the ^11^B{^1^H} NMR of **2**. The mass spectra of **1** and **2** showed peaks at *m*/*z* 1234.5158 and *m*/*z* 1312.4354, respectively. Further, single crystal X-ray diffraction analyses were carried out on suitable crystals, **1** and **2**, to confirm the spectroscopic assignments and to conclude their structures without any ambiguity.

The single crystal X-ray analyses revealed compounds **1** and **2** as the trimetallic species [(NbCp*)_3_(*μ*_3_-Se)_3_(BH)(*μ*-Se)_3_] (Figure 1a) and [(NbCp*)_3_(*μ*_3_-Se)_3_(*μ*-Se)_3_(BH)(*μ*-Se)] (**2**) (Figure 1b), respectively. Both the trimetallic species consist of an equilateral Nb_3_ triangular core. The Nb-Nb bond distances in **1** (3.270(2), 3.271(2) and 3.272(3) Å), and in **2** (3.254(5), and 3.307(4) Å) are comparable. They are slightly longer than those reported in other niobaboranes and niobaheteroboranes [50,53,55]. The internal angles (∠Nb-Nb-Nb) of the triangular framework in **1** (59.96(5)°, 59.99(5)° and 60.05(5)°), and **2** (58.94(10)°, and 60.53(5)°) are close to 60°. Six monoselenide atoms bridge all the edges of the Nb_3_-triangle in compounds **1** and **2**. In the case of **1**, the three selenide units are capped by a {BH} unit. Therefore, a selenaborate {Se_3_BH}^−^ ligand coordinates the triangular tri-niobium framework from the top in **1**. The B-Se bond distances of 2.095(18), 2.101(18), and 2.107(18) Å in **1** are comparable with those measured in the reported metallaheteroboranes [52,54]. Alternatively, the core geometry of **1** can also be viewed as a cubane-like cluster with one missing vertex, a niobium-selenium analogue of the trimetallic species [(TaCp*)_3_(*μ*_3_-S)_3_(BH)(*μ*-S)_3_] [51]. As three Nb-Nb bonds are present and one metal vertex is missing in **1**, the cluster valence electron count for trimetallic **1** must be {80 − 18 − (3 × 2)} = 56; however, the actual electron count of **1** is {3(NbCp*) + 3(*µ*_3_-Se) + 3(*µ*-Se) + (BH)} = {30 + 12 + 6 + 2} = 50. Therefore, cluster **1** is a hypo-electronic cluster. On the other hand, two of the bridging Se atoms of **2** are directly bonded to the boron atom, while the other selenium atom (Se5) is bridged between B1 and Se4 atoms in **2**. The B-Se and Se-Se bond distances of **2** are comparable with other reported B-Se and Se-Se single bonds. As the core geometry of tri-niobium species **2** can be generated by incorporating a selenium atom in between one of the B-Se bonds of cubane-type cluster **1**, trimetallic species **2** can be viewed as a homocubane-like cluster with one missing vertex, a selenium analogue of the trimetallic species [(NbCp*)_3_(*μ*_3_-S)_3_(*μ*-S)_3_(BH)(*μ*-S)] (**I**) [53]. On the other hand, a *C*_3_ axis passes through the BH unit and the midpoint of the Nb_3_ triangle of **1**, which makes compound **1** a symmetric cluster, and its spectroscopic data positively corroborates with the X-ray structure. In addition, in line with the spectroscopic data, the solid-state structure of **2** exhibits a symmetry plane that passes through the Nb1′, B1, Se5, Se4 and the centre of the Nb1-Nb2 bond.

### 2.2. Reactivity of [Cp*NbCl_4_] with Li[BH_2_S_3_]

In order to isolate the sulfur analogue of **1**, we performed a thermolysis reaction of [Cp*NbCl_4_] with four equivalents of [LiBH_2_S_3_], for a prolonged time, which led to the formation of a yellow solid, as well as the known trimetallic homocubane-like cluster [(NbCp*)_3_(*μ*_3_-S)_3_(*μ*-S)_3_(BH)(*μ*-S)] (**I**) (Figure 2) [53]. The ^1^H NMR spectrum of the yellow compound showed two ^1^H resonances in a 2:1 ratio at *δ* = 2.10 and 2.03 ppm, which suggests the presence of two different Cp* environments. The presence of two different Cp* environments was further confirmed by the ^13^C{^1^H} spectrum. Moreover, the ^1^H NMR spectrum showed a peak at *δ* = 3.54 ppm that suggested the presence of a terminal B-*H* proton. Further, the presence of the terminal B-*H* proton was confirmed by IR spectroscopy. In addition, the ^11^B{^1^H} NMR spectrum of the yellow compound showed a peak at *δ* = 3.8 ppm. The mass spectrum showed molecular ion peaks at *m*/*z* 876.9071 and *m*/*z* 908.8798. However, these spectroscopic data were puzzling for forming a conclusion about the molecular structure. Thus, we have carried out single crystal X-ray diffraction analyses on a suitable yellow crystal to elucidate the solid-state structure.

Surprisingly, the single crystal X-ray analyses revealed the yellow crystal as a co-crystal of two inseparable solids (**3** and **4**). As shown in Figure 2, compounds **3** and **4** are the trimetallic species [(NbCp*)_3_(*μ*-S)_4_{*μ*-S_2_(BH)}] and [(Cp*Nb)_3_(*μ*-S)_4_{*μ*-S_2_(S)}], respectively. Both species **3** and **4** have an equilateral Nb_3_ triangular core geometry. The Nb-Nb bond distances of 3.242 and 3.006 Å in **3** and **4**, respectively, are slightly shorter, especially the latter, than those in **1** and **2**. The internal angles (∠Nb-Nb-Nb) of the triangular framework in **3** and **4** (62.38° and 55.25°) are also close to 60°. All the edges of the Nb_3_-triangle in **3** and **4** are bridged by six mono-sulfide atoms. Further, two of these mono-sulfide atoms in **3** and **4** are bridged by {BH} and another by mono-sulfide units, respectively. Compound **4** is the Cp* analogue of [(NbCp^+^)_3_(*μ*-S)_4_{*μ*-S_2_(S)}] (Cp^+^ = *ɳ*^5^-C_5_Me_4_Et) reported earlier [56]. On the other hand, the thioborane unit {BS_2_H} that coordinates two of the edges of the triangular tri-niobium framework from the top in **3** is quite unique. The B-S bond distances of 2.18(3) and 2.23(3) Å in **3** are slightly longer than those reported in the other metallaheteroboranes [38]. Interestingly, the thioborane unit also forms a tetratomic nonplanar metallathioboron cycle (Nb2-S1-B1-S4′) with one of the Nb atoms. The X-ray structures of **3** and **4** corroborate well with their combined spectroscopic data. For example, ^11^B{^1^H} chemical shifts at *δ* = 3.8 ppm and ^1^H chemical shift at *δ* = 3.54 ppm correspond to the {BH} unit of **3**. On the other hand, the ^1^H chemical shifts correspond to the Cp* ligands of **3** and **4**; these should each appear as a 2:1 ratio for three Cp* ligands, as the symmetry plane passes through an Nb2 vector and the centre of the Nb1-Nb2′ bond in **3**, and an Nb1 vector and the centre of the Nb2-Nb2′ bond in **4**. However, in the combined ^1^H NMR spectrum of **3** and **4**, these peaks for Cp* ligands accidentally overlapped and appeared in a 4:2; i.e., 2:1 ratio.

### 2.3. Electronic Structural and Bonding Analysis of ***1***–***4***

To gain some insight into the bonding and electronic structures of **1**–**4**, we performed density functional theory (DFT) calculations at the BP86/def2-svp level of the theory [57,58]. Interestingly, the molecular orbital (MO) analyses showed that the HOMOs of **1**–**4** are mainly made of *d*-orbitals of the three Nb atoms, which overlap with each other and fully delocalize over the whole of the triangular trimetallic framework (Figure 3), illustrating 2-electron-3-centre bonding. Further, NBO [59,60,61] analyses were carried out to investigate more precisely the bonding in the Nb_3_ triangle. The WBIs [62] of 0.362, and 0.365 in **1**; 0.324, and 0.381 in **2**; 0.355, and 0.467 in **3**; and 0.331, and 0.532 in **4** further support bonding interactions between Nb atoms within these species to form a Nb_3_ triangular skeleton. As shown in [Fig molecules-27-07473-ch001], these types of trimetallic molecules show σ or π-aromaticity. This prompted us to probe the delocalized bonding in the Nb_3_ core of **1**–**4**. Thus, we employed the nucleus independent chemical shift (NICS) approach [21,22] for assessing the aromaticity of clusters **1**–**4**. The NICS(0) (the negative of the isotropic magnetic shielding at the centre of the Nb_3_ ring) for **1**–**4** were found to be −16.8, −20.3, −17.7, and −21.0 ppm, respectively, with negative numbers indicating aromaticity. In addition, the NICS(0)*_zz_* values of **1**–**4**, −48.1, −46.8, −45.7, and −47.5, respectively, quantify the out of the plane aromaticity. The NICS(0.5), NICS(−0.5), NICS(1), and NICS (−1) values computed for **1**–**4** are also negative, which show a ring current, indicating the presence of induced diatropic ring currents, which typically suggests three-dimensional aromaticity (Appendix A). Therefore, the existence of an aromatic system plays a key role in the stabilization of **1**–**4**.

The triangular trimetallic skeleton of **1**–**4** further captured borate, borane, and chalcogen units. Therefore, we further analyzed the way they bind together. An MO analysis of **1** showed significant orbital overlap interaction between the triangular Nb_3_ framework and the selenaborate {Se_3_BH}^−^ ligand, which is depicted in HOMO-3, HOMO-6, HOMO-7, and HOMO-8 (Figure 4a and Appendix A). Further, an NBO analysis depicted the strong B-Se bonding interaction within the selenaborate {Se_3_BH}^−^ ligand in **1**, which is further supported by WBI (0.887, 0.887, and 0.883) for B-Se interactions. With the exception of the bonding scenario along B1-Se7 and Se7-Se5, similar types of bonding interactions were observed in the MO (Figure 4d and Appendix A) and NBO (Figure 4e) analyses of **2**. WBIs of 1.064 and 0.909 support the strong bonding interaction between B1-Se and Se7-Se5, respectively. Further, a natural charge analysis of **1** and **2** showed that the three Nb atoms and the B atom act as acceptors, whereas the Se atoms act as donors (Appendix A). On the other hand, the MO analyses of **3** and **4** showed significant orbital overlap interaction in the triangular Nb_3_ framework and the directly connected six S atoms (Appendix A). In addition, substantial orbital overlap interactions among the p-orbitals of two of these directly metal-connected chalcogens and {BH} unit of **3,** and mono-sulfur unit of **4**, are depicted in HOMO-23 (**3**), and HOMO-8 (**4**), respectively (4(g) and Appendix A). In addition, an NBO analysis also illustrates the S-B bonding interaction in **3** (Figure 4h), which is also supported by higher WBIs (Appendix A). Additionally, the Laplacian plot of electron density [63,64,65] showed areas of charge concentration, BCPs, and bond paths between B-Se in **1** (Figure 4c), B-Se and Se-Se in **3** (Figure 4f), S-B in **3** (Figure 4i), and S and S in **4** (Appendix A). Finally, the HOMO-LUMO energy gap in **2** (1.224 eV) is relatively smaller than those in **1** (1.296 eV), **3** (1.467 eV), and **4** (1.402 eV).

## 3. Materials and Methods

All manipulations were carried out under an inert atmosphere of argon, either inside a glove box or by making use of standard Schlenk line techniques. All of the solvents were dried and distilled before use, employing standard literature procedures. [Cp*NbCl_4_] [66], Li[BH_2_E_3_] (E = S and Se) [67,68], and the external reference used for analyzing ^11^B{^1^H} NMR, [Bu_4_N(B_3_H_8_)], were all prepared in accordance with reported procedures [69]. [LiBH_4_∙THF], S powder, and Se powder (Merck KGaA, Darmstadt, Germany) were employed as received. The reaction mixtures were separated by performing thin layer chromatography using aluminum-supported silica gel TLC plates (Merck KGaA, Darmstadt, Germany) of 250 µm diameter. A 500 MHz Bruker FT-NMR spectrometer (Bruker, Billerica, MA, USA) was used to record ^1^H, ^11^B{^1^H}, and ^13^C NMR spectra. The residual solvent protons were taken as a reference (CDCl_3_, *δ* = 7.26 ppm), whereas a [D_6_] benzene solution of [Bu_4_N(B_3_H_8_)], taken in a sealed tube, was used as an external reference (*δ*_B_ = −30.07 ppm) for analyzing ^11^B{^1^H} NMR spectra. A Bruker MicroTOF-II mass spectrometer (Bruker Daltonics, Bremen, Germany) was used to record ESI mass spectra. IR spectra were recorded with a JASCO FT/IR-1400 spectrometer (JASCO, Easton, PA, USA). It is important to note that elemental analysis of these compounds could not be performed due to low yields and higher sensitivity.

### 3.1. Synthesis of ***1*** and ***2***

A suspension of Cp*NbCl_4_ (0.20 g, 0.54 mmol) in 20 mL of toluene was made in a flame-dried Schlenk tube. A freshly prepared solution of Li[BH_2_Se_3_] (2.16 mmol) in THF was added to the toluene suspension. The reaction mixture was stirred and heated at 80 ℃ for 48 h. The solvent was dried by applying a vacuum, and the residue was extracted into an *n*-hexane/CH_2_Cl_2_ mixture (80:20 *v*/*v*) and passed through Celite. After removing the volatiles by applying a vacuum, chromatographic separation of the residue was carried out using TLC plates. Elution with a *n*-hexane/CH_2_Cl_2_ mixture (70:30 *v*/*v*) formed brown solid **1** (0.023 g, 11% yield) and green solid **2** (0.027 g, 12% yield).

Spectroscopic data of **1**: MS (ESI^+^): *m*/*z* calculated for C_32_NH_48_Nb_3_Se_6_BNa^+^ [(M − H) + Na + CH_3_CN]^+^: 1234.6009, found: 1234.5158. ^11^B{^1^H} NMR (160 MHz, CDCl_3_, 22 °C): *δ* = −8.3 ppm; ^1^H NMR (500 MHz, CDCl_3_, 22 °C): *δ* = 2.21 (s, 45 H; 3 × Cp*), 3.87 ppm (b, B-*H*_t_); ^13^C{^1^H} NMR (125 MHz, CDCl_3_, 22 °C): *δ* = 14.7 (C_5_*Me*_5_), 117.9 ppm (*C*_5_Me_5_); IR (CH_2_Cl_2_, cm^−1^): 2515 (B-*H*_t_).

Spectroscopic data of **2**: MS (ESI^+^): *m/z* calculated for C_30_H_47_Nb_3_Se_7_NaKH^+^ [M + Na + K + H]^+^ 1312.4714, found: 1312.4354. ^11^B{^1^H} NMR (160 MHz, CDCl_3_, 22 °C): *δ* = −6.4 ppm ^1^H NMR (500 MHz, CDCl_3_, 22 °C): *δ* = 2.18 (s, 15H; 1 × Cp*), 2.18 ppm (s, 30H; 2 × Cp*), 3.65 ppm (b, B-*H*_t_); ^13^C{^1^H} NMR (125 MHz, CDCl_3_, 22 °C): *δ* = 14.3 (C_5_*Me*_5_), 14.8 (C_5_*Me*_5_), 116.9 (*C_5_*Me_5_), 117.8 ppm (*C*_5_Me_5_); IR (CH_2_Cl_2_, cm^−1^): 2489 (B-*H*_t_).

### 3.2. Synthesis of ***3*** and ***4***

Under similar experimental conditions, a reaction of Li[BH_2_S_3_] (2.16 mmol) with Cp*NbCl_4_ (0.20 g, 0.54 mmol) was carried out. The solvent was dried by applying a vacuum, and the residue was extracted into an *n*-hexane/CH_2_Cl_2_ mixture (80:20 *v*/*v*) and passed through Celite. After removing the volatiles by applying a vacuum, chromatographic separation of the residue was carried out using TLC plates. Elution with *n*-hexane/CH_2_Cl_2_ mixture (70:30 *v*/*v*) formed a mixture of **3** and **4** as a yellow solid (0.026 g, 16% yield) along with a known orange solid [(NbCp*)_3_(*μ*_3_-S)_3_(*μ*-S)_3_(BH)(*μ*-S)] [53] (0.015 g, 9% yield).

Combined spectroscopic data of **3** and **4**: MS (ESI^+^): *m*/*z* calculated for C_30_H_46_Nb_3_S_6_^+^ [M − BH + H]^+^: 876.9115, found: 876.9071; *m*/*z* calculated for C_30_H_46_Nb_3_S_7_^+^ [M + H]^+^: 908.8836, found: 908.8798. ^11^B{^1^H} NMR (160 MHz, CDCl_3_, 22 °C): *δ* = 3.8 ppm; ^1^H NMR (500 MHz, CDCl_3_, 22 °C): *δ* = 2.03 (s, 30 H; 2 × Cp*), 2.10 (s, 60 H; 4 × Cp*), 3.54 ppm (b, B-*H*_t_); ^13^C{^1^H} NMR (125 MHz, CDCl_3_, 22 °C): *δ* = 12.9 (C_5_*Me*_5_), 13.1 (C_5_*Me*_5_), 116.9 (*C*_5_Me_5_), 119.0 (*C*_5_Me_5_); IR (CH_2_Cl_2_, cm^−1^): 2472 (B-*H*_t_).

### 3.3. X-ray Structure Determination

A Bruker AXS Kappa APEX-II CCD diffractometer (Bruker AXS Inc., Madison, WI, USA) bearing a graphite monochromated MoK_α_ (λ = 0.71073 Å), radiated at 296 K, was used to collect and integrate the crystal data of compounds **1**–**5**. Heavy atom methods were employed to solve the structures by making use of SHELXS-97 or SIR92, and structure refinement was achieved using SHELXL-2018/3 [70,71,72]. Structures of the molecules were drawn with Olex2 [73]. Note that all atoms of the disordered component of **1** were found to have shifted by the same vector (0.3958, −0.2374, 0.0078) as the corresponding atoms of the main molecule. This showed the disorder component to be a translation of the main molecule through a distance of 6.631 Å in the direction of the vector (0.3958, −0.2374, 0.0078). The Cp* and the boron atoms of the main molecule were moved through this vector in a separate job. These atoms’ cards were then copied and put in the main res file, relabeled and assigned the occupancies −21.0 The Cp* and boron atoms became correctly attached to the disorder component. The molecules were refined with suitable restraints. As the scattering strength of the carbon atoms of the minor disordered component is only 6/20 = 0.3 e, the restraints were needed for refinement. Note that multiple recrystallizations of **2** from various solvents provided extremely thin crystals, which diffracted poorly. Thus, *R*_int_ is high for **2**. However, proper analysis of the X-ray data provided conclusive proof of the structure, which also positively corroborated with the other spectroscopic evidence. In addition, the atoms Se5 and B1 had 50% occupancy only and of the bonds Se5-B1 and Se5′-B1′, any one of them only existed in a given molecule. Compound **2** crystallized along with the CHCl_3_ solvent molecule. Compounds **3** and **4** are a 50:50 mixture of a yellow co-crystal and crystallized along with two CH_2_Cl_2_ solvent molecules, whereas The Cambridge Crystallographic Data Centre has been provided with the crystallographic data of the compounds with supplementary publication no. CCDC-2203605 (**1**), 2203607 (**2**), and 2203609 (**3** and **4**).

Crystal data of **1**: [C_30_H_46_BNb_3_Se_6_], *M*_r_ = 1169.97, triclinic, *P*$\bar{1}$, *a* = 11.984(2) Å, *b* = 11.986(2) Å, *c* = 15.488(3) Å, *α* = 98.184(8)°, *β* = 98.597(9)°, *γ* = 119.817(7)°, *V* = 1846.7(6) Å^3^, *Z* = 2, *ρ*_calc_ = 2.104 g/cm^3^, *µ* = 6.848 mm^−1^, *F*(000) = 1116.0, *R*_1_ = 0.0884, *wR*_2_ = 0.2140, 5681 independent reflections, [2θ ≤ 48.792] and 727 parameters.

Crystal data of **2**: 2[C_30_H_46_B_1_Nb_3_Se_7_] + [CHCl_3_], *M*_r_ = 2617.22, orthorhombic, *Cmc*2_1_, *a* = 15.792(3) Å, *b* = 16.816(3) Å, *c* = 16.237(2) Å, *α* = 90°, *β* = 90°, *γ* = 90°, *V* = 4312.0(12) Å^3^, *Z* = 2, *ρ*_calc_ = 2.016 g/cm^3^, *µ* = 6.805 mm^−1^, *F*(000) = 2484.0, *R*_1_ = 0.0739, *wR*_2_ = 0.1494, 3944 independent reflections, [2θ ≤ 49.998] and 235 parameters.

Crystal data of **3** and **4**: [C_30_H_46_BS_6_Nb_3_] + [C_30_H_45_S_7_Nb_3_] + 2 [CH_2_Cl_2_], *M*_r_ = 1090.48, orthorhombic, *Pnma*, *a* = 16.903(3) Å, *b* = 16.437(3) Å, *c* = 15.215(3) Å, *α* = *β* = *γ* = 90°, *V* = 4227.3(14) Å^3^, *Z* = 4, *ρ*_calc_ = 1.713 g/cm^3^, *µ* = 1.422 mm^−1^, *F*(000) = 2200, *R*_1_ = 0.0546, *w*R_2_ = 0.1606, 3857 independent reflections, [2θ ≤ 49.998] and 232 parameters.

### 3.4. Computational Details

All molecules were fully optimized using the BP86 functional [57,58], in conjunction with a def2-svp basis set using the *Gaussian 09* program (Gaussian, Wallingford, CT, USA) [74]. All compounds were fully optimized in gaseous state using their X-ray crystallographic structures. The calculations were performed with the Cp analogues, instead of Cp*, to save computing time. Note that the B1···S4 distance of **3** was frozen and optimized. Further, frequency calculations of **3** confirmed the absence of any imaginary frequency. NBO analyses were carried out with the NBO partitioning scheme [59,60,61] as employed in *Gaussian 09*. Wiberg bond indexes (WBI) [62] were obtained from the NBO analysis. QTAIM analyses [63,64,65] were performed utilizing the *Multiwfn* V.3.6 package (Multiwfn, Beijing, China) [75]. The aromaticity of the compounds was evaluated by calculating the Nucleus Independent Chemical Shift (NICS) indices [21,22] on the optimized geometries, at the same level of theory, by using the GIAO method implemented in *Gaussian*. All the optimized structures and orbital graphics were produced using *Gaussview* [76] and *Chemcraft* [77].

## 4. Conclusions

In summary, we have synthesized and structurally characterized novel species containing a 2-electron aromatic tri-niobium skeleton, which further stabilizes different types of main group units, such as borate, borane, and mono/di-chalcogenide units. The common cluster core of all these novel species is Nb_3_E_6_ (E = S, Se). Theoretical calculations revealed the participation of Nb d-orbitals to form an aromatic Nb_3_ framework. In addition, the pivotal role of the aromatic Nb_3_ skeleton in the stabilization of borate, borane, and mono/di-chalcogenide units is revealed. Additional exploration to evaluate the scope for isolation of the other main group units, utilizing the trimetallic skeleton of group 4 and 5 transition metals, is currently underway.

## Data Availability

Supporting data reported can be found as Appendix A.

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
