# Peer review of "Trimetallic Chalcogenide Species: Synthesis, Structures, and Bonding"

_molecules, 2022, doi:10.3390/molecules27217473_

Round 1
Reviewer 1 Report
The manuscript describes the synthesis and characterisation of trimetallic polychalcogenides.
The manuscript is well written, the science is sound. As such, I see no reason not to publish this manuscript, however, I would appreciate authors' reconsideration/revise of the following points before the final version is published.
1. The titleis too large to represent the reported compounds, especially on "polychalcogenide", it includes S, Se, Te. Moreover, the aromaticity is too much to the experimental results shown in the manuscript.
2. Related the aromaticity, do you have any data on 77Se NMR? Anyhow, The word "aromaticity" in the title is too excessive.
3. X-ray crystallography of 2+3 and 4+5 was less favorable; 2+3 was relatively acceptable because Se was spontaneously disproportionated or extruded from the Se2 bridge in the compound, while 4+5 with/without the B-H bridge was probably a problem due to insufficient purification of the target compound(s).
4. X-ray crystallography of 1 is also somewhat ambiguous. Is this the result of "disorder"? Is there any possibility of a result of combined crystals of large and small one or non-merohedral twinning? Although the chemistry is fine, I do NOT like many-many DFIX, SIMU, FLAT, ISOR for one molecule at the same time, you may need Re-take or recrystallization with the other conditions for compound 1.
********Thanx.
Author Response
The manuscript describes the synthesis and characterisation of trimetallic polychalcogenides. The manuscript is well written, the science is sound. As such, I see no reason not to publish this manuscript, however, I would appreciate authors' reconsideration/revise of the following points before the final version is published.
Response: We appreciate the reviewer's positive feedback. We are glad to address reviewer’s comments and remarks.
- The title is too large to represent the reported compounds, especially on "polychalcogenide", it includes S, Se, Te. Moreover, the aromaticity is too much to the experimental results shown in the manuscript.
Response: We have modified the title accordingly.
- Related the aromaticity, do you have any data on 77Se NMR? Anyhow, the word "aromaticity" in the title is too excessive.
Response: We thank the reviewer for the thoughtful suggestion. We have tried to record 77Se NMR of compounds 1 and 2. However, we did not observe any chemical shift due to low yields. We hope that the reviewer will understand the situation. We have removed the word ‘aromaticity’ form the title as per the suggestion.
- X-ray crystallography of 2+3 and 4+5 was less favorable; 2+3 was relatively acceptable because Se was spontaneously disproportionated or extruded from the Se2 bridge in the compound, while 4+5 with/without the B-H bridge was probably a problem due to insufficient purification of the target compound(s).
Response: We thank the reviewer for the careful evaluation. We have rechecked and freshly refined the X-ray data. The crystal of 2 (earlier compounds 2+3) was poorly diffracting at higher Bragg angles. Re-crystallization and fresh data collection did not improve the situation unfortunately. However, the second reviewer’s suggestion about the large voids present in the structure of 2 was found to be very valid and careful inspection of difference map has revealed the compound to be having a different structure [(NbCp*)3(μ3-Se)3(μ-Se)3(BH)(μ-Se)] (2), which is also supported by re-recorded spectroscopic data of the pure compound. The revised cif with the new structure is submitted. Also, we understand the reviewer’s concern regarding the purity of compounds 3+4 (earlier compounds 4+5). The crystal under investigation shows a 1:1 ratio of the components, which is also supported by other analysis techniques such as NMR. The electron density peak near S5 at a distance of 1.28 Å was assigned as B1 (S5 and B1 are mutually exclusive). The refinement parameters and thermal parameters of the atom B1 indicated that it can have an occupancy of 0.5 in the asymmetric unit of the crystal lattice.
Also, we have tried our best to purify compounds 3 and 4 using the chromatography technique; however, we were unsuccessful due to having same polarities. The identity and purity of these compounds can be clearly understood from Cp* region peaks that appeared at δ = 2.10 and 2.03 ppm in 2:1 (or 4:2) ratio in 1H NMR. The number of peaks appeared in this region corresponds to Cp* ligands of compounds 3 and 4 only, which are accidentally overlapped. Also, the boron-containing compound (3) shows only one 11B chemical shift at δ = 3.8 ppm; therefore, no other boron compound is present. We hope the reviewer will understand our situation regarding the difficulty in the isolation of 3 and 4.
- X-ray crystallography of 1 is also somewhat ambiguous. Is this the result of "disorder"? Is there any possibility of a result of combined crystals of large and small one or non-merohedral twinning? Although the chemistry is fine, I do NOT like many-many DFIX, SIMU, FLAT, ISOR for one molecule at the same time, you may need Re-take or recrystallization with the other conditions for compound 1.
Response: We thank the reviewer for his (her) suggestions. The crystal data was tested for non-merohedral twinning during data collection and afterward during structure solution. It was found that no twinning was present. If we inspect the packing of the unit cell with major component of disorder alone, it can be seen that the molecule with its inversion pair aligns nearly in the [1 1 1] direction leaving large void regions in the [1 1 -1] direction. The minor disorder component (occupancy 4.5%) and its inversion occupy these void regions. The minor component might be forming micro-domains in the crystal without changes in the crystallographic axis directions. The minor component of disorder was not arbitrarily built up by constraints and restraints. These constraints were put during the structure development and completion stages. The structure was freshly refined after removing most of the constraints and restraints and fresh cif is submitted.
Reviewer 2 Report
The manuscript reports the synthesis and characterization of five trimetallic polychalcogenide complexes which are shown to display aromaticity. The work has significance. I am not expert in the type of theoretical analysis performed, and so cannot comment on that. I do have concerns regarding some of the work. It is unfortunate that there are co-crystals for two pairs of the compounds, and this has consequences for X-ray diffraction studies and characterization by other techniques.
The characterization of complex 1 is fine, although the structural data is not ideal, and I think it may be incorrect. As presented there is a lot of space in the unit cell (ca. 17% - occupied by the molecules of the alternative site). It looks as though there might be a plane of symmetry through the molecule, and so I suspect that the space group is not P-1, but of higher symmetry, and there is twinning that appears to give what is modelled as the minor site. This is probably is a consequence of the diffraction being performed at room temperature. Nonetheless it would have been useful to obtain the elemental analysis to support the characterization for the bulk sample, especially as the NMR spectra are so simple.
For co-crystal II (complexes 4 and 5) I agree with the identities of the species. However, I do have some questions. The authors have set the ratio as 1:1 for the structural analysis, and assumed this is the case in the bulk sample. Firstly, did they try any other ratios for the structural analysis? Secondly, is it possible that the NMR spectra are consistent with other ratios. Again elemental analysis would have been helpful.
The characterization of complexes 2 and 3 is more problematic. There is no mass spectral data indicating the presence of complex 2, and the NMR spectra are more consistent with either 2 or 3, rather than a mixture, for which coincidence of both 1H and 13C resonances is required. The X-ray diffraction data is of very poor quality (Rint = 22.24%, well above what is considered satisfactory), and the overall refinement parameters are consequently poor. Furthermore, there are disturbingly large voids in the structure, accounting for 5% of the volume, that could accommodate a molecule such as water. The voids would be larger when they are adjacent to complex 2, because of the space afforded by the loss of two selenium atoms. I feel that there is insufficient data to convince me of the identity of I as a 1:1 mixture of 2 and 3. The crystal data is not of good enough quality to provide a definitive characterization of the complexes, and the NMR and mass spectral data is not sufficient to confirm the identities of both 2 and 3. Elemental analysis would be immensely supportive for the characterization.
Consequently, major revision is required. I strongly recommend that the authors make attempts to obtain elemental analyses for all three species. If it is not possible to do so, then a comment regarding this and an explanation should be included in the manuscript. If possible, the X-ray diffraction of I and II should be performed at lower temperature and for I with a better quality crystal. Currently, they are not of sufficient quality to allow publication.
Some other problems need to be addressed:
page 2, line 69: the last sentence of the paragraph needs revision.
page 5, line 168: the Cp* resonances are coincident not collapsed.
Some chemical shifts are given to 3 decimal places (e.g. page 2 line 77). This is overly optimistic. 2 Decimal places are all that can be justified for 1H NMR.
Check the resolution of the IR spectrometer. I think that it is 0.4 cm-1, and so quoting the frequency to 2 decimal places is not justified.
Descriptions of the treatment of the disorder in the crystal structures should be given in the experimental section.
Author Response
The manuscript reports the synthesis and characterization of five trimetallic polychalcogenide complexes which are shown to display aromaticity. The work has significance. I am not expert in the type of theoretical analysis performed, and so cannot comment on that. I do have concerns regarding some of the work. It is unfortunate that there are co-crystals for two pairs of the compounds, and this has consequences for X-ray diffraction studies and characterization by other techniques.
Response: We are grateful to this reviewer for precisely reviewing this manuscript with insightful comments and suggestions.
- The characterization of complex 1 is fine, although the structural data is not ideal, and I think it may be incorrect. As presented, there is a lot of space in the unit cell (ca. 17% - occupied by the molecules of the alternative site). It looks as though there might be a plane of symmetry through the molecule, and so I suspect that the space group is not P-1, but of higher symmetry, and there is twinning that appears to give what is modelled as the minor site. This is probably is a consequence of the diffraction being performed at room temperature. Nonetheless it would have been useful to obtain the elemental analysis to support the characterization for the bulk sample, especially as the NMR spectra are so simple.
Response: We thank the reviewer for the careful evaluation. The ab plane of the lattice is near hexagonal, a and bangles are nearly equal but quite different from 90 degrees. The unit cell could be transformed to a near C-centered unit cell but showed no semblance of structure with any monoclinic space group. The space group is P-1 itself. The crystal data was tested for non-merohedral twinning during data collection and afterward during structure solution. It was found that no twinning was present. If we inspect the packing of the unit cell with major component of disorder alone, it can be seen that the molecule and its inversion pair align nearly in [1 1 1] direction leaving large void regions in the [1 1 -1] direction. The minor disorder component (occupancy 4.5%) and its inversion occupy these void regions. The minor component might be forming micro-domains in the crystal without changes in the crystallographic axis direction. The minor component of disorder was not arbitrarily built up by constraints and restraints. These constraints were put during the structure development and completion stages. The structure was freshly refined after removing most of the constraints and restraints and fresh cif is submitted. The procedures used for locating the minor component (4.5% occupancy) of disorder of compound 1 are as follows.
First, the main molecule is completed, and 20 difference peaks are generated.
Step-1: The shelxl res file with 20 differences.
Step-2: Deleted all atoms from the res file leaving diff peaks alone.
Step-3: We have assigned peaks Q1, Q2, Q4 as Nb and Q3, Q5, Q6, Q7, Q9, Q11 as Se atoms.
Step-4: Then we saved the atoms as trial.res.
Step-5: After that, we opened trial.res in SXGRAPH and did Grow Fragment. The fragment grows as Nb-Se core.
Step-6: Then, atom lines from trial.ins is copied and added to main molecule res file as disorder components in part 2 with site occupancy of -21.00000. The Nb and Se atoms of the disordered component are relabeled to follow the same labelling sequence of the main molecule.
Step-7: All atoms of the disordered component were found to have shifted by the same vector (0.3958, -0.2374, 0.0078) from the corresponding atoms of the main molecule. This showed the disorder component to be just a translation of the main molecule through a distance of 6.631 angstroms in the direction of the vector (0.3958, -0.2374, 0.0078). The Cp* and the boron atoms of the main molecule is moved through this vector in a separate job. These atoms cards were then copied and put in the main res file, relabeled and assigned the occupancies -21.00000. The Cp* and boron atoms got correctly attached to the disorder component. The molecules were refined with suitable restraints. As the scattering strength of the carbon atoms of the minor disordered component is only 6/20= 0.3 e, the restraints were needed for refinement.
Also, we completely understand the importance of elemental analysis on the bulk sample and the reviewer’s concern in this regard. We have tried the elemental analysis for compound 1; however, the results were unsatisfactory, and this may be due to the high sensitivity of the compound or the operator’s failure. Hence, at this stage, we are unable to provide the elemental analysis data. Hope the reviewer will understand our situation.
- 2. For co-crystal II (complexes 4 and 5) I agree with the identities of the species. However, I do have some questions. The authors have set the ratio as 1:1 for the structural analysis, and assumed this is the case in the bulk sample. Firstly, did they try any other ratios for the structural analysis? Secondly, is it possible that the NMR spectra are consistent with other ratios. Again elemental analysis would have been helpful.
Response: We thank the reviewer for the suggestions. We have rechecked the X-ray data of complexes 3 and 4 (earlier complexes 4 and 5). The crystal under investigation shows a 1:1 ratio of the components, which is also supported by other analysis techniques such as NMR. The electron density peak near S5 at a distance of 1.28 Å was assigned as B1 (S5 and B1 are mutually exclusive). The refinement parameters and thermal parameters of the atom B1 indicated that it can have an occupancy of 0.5 in the asymmetric unit of the crystal lattice. We have rechecked the 1H, 11B{1H}, and 13C{1H} NMR data of 3 and 4; however, they are only consistent with the presented ratio in the manuscript. We understand the importance of elemental analysis on the bulk sample and the reviewer’s concern in this regard. However, due to the low yields and high sensitivity, we could not perform the elemental analysis of compounds 3 and 4. Hence, at this stage, we are unable to provide the elemental analysis data. Hope the reviewer will understand our situation.
- The characterization of complexes 2 and 3 is more problematic. There is no mass spectral data indicating the presence of complex 2, and the NMR spectra are more consistent with either 2 or 3, rather than a mixture, for which coincidence of both 1H and 13C resonances is required. The X-ray diffraction data is of very poor quality (Rint = 22.24%, well above what is considered satisfactory), and the overall refinement parameters are consequently poor. Furthermore, there are disturbingly large voids in the structure, accounting for 5% of the volume, that could accommodate a molecule such as water. The voids would be larger when they are adjacent to complex 2, because of the space afforded by the loss of two selenium atoms. I feel that there is insufficient data to convince me of the identity of I as a 1:1 mixture of 2 and 3. The crystal data is not of good enough quality to provide a definitive characterization of the complexes, and the NMR and mass spectral data is not sufficient to confirm the identities of both 2 and 3. Elemental analysis would be immensely supportive for the characterization.
Response: We completely understand the reviewer's concern. Also, we are very much thankful for the significant suggestion. The crystal was poorly diffracting at higher Bragg angles. Too many weak reflections at higher Bragg angles have caused high Rint value. Re-crystallization and fresh data collection did not improve the situation. The referee’s suggestion about the large voids present in the structure was found to be very valid and careful inspection of difference map has revealed the compound to be having a different structure [(NbCp*)3(μ3-Se)3(μ-Se)3(BH)(μ-Se)] (2), which is also supported by re-recorded spectroscopic data of the pure compound. The cif with the new structure is submitted. As we mentioned earlier, we completely understand the importance of elemental analysis on the bulk sample; however, due to the low yields and high sensitivity, we could not perform the elemental analysis of compound 2. Hence, at this stage, we are unable to provide the elemental analysis data. Hope the reviewer will understand our situation.
- Consequently, major revision is required. I strongly recommend that the authors make attempts to obtain elemental analyses for all three species. If it is not possible to do so, then a comment regarding this and an explanation should be included in the manuscript. If possible, the X-ray diffraction of I and II should be performed at lower temperature and for I with a better quality crystal. Currently, they are not of sufficient quality to allow publication.
Response: As mentioned earlier, we completely understand the importance of elemental analysis on the bulk sample and the reviewer’s concern in this regard. Due to the low yields and high sensitivity, we could not perform the elemental analysis of compounds 2-4. We have tried the elemental analysis for compound 1; however, the results were unsatisfactory, and this may be due to high sensitivity or operator’s failure. Hence, at this stage, we are unable to provide the elemental analysis data for all the compounds. Hope the reviewer will understand our situation. We have added a comment regarding this and an explanation in the manuscript. Also, as per the suggestion, we have freshly refined the X-ray data, which really helped us to determine the structure properly.
Some other problems need to be addressed:
- page 2, line 69: the last sentence of the paragraph needs revision.
Response: We have now rewritten the sentence in the revised manuscript.
- page 5, line 168: the Cp* resonances are coincident not collapsed.
Response: We thank the reviewer for the careful evaluation. We have now corrected it in the revised manuscript.
- Some chemical shifts are given to 3 decimal places (e.g. page 2 line 77). This is overly optimistic. 2 Decimal places are all that can be justified for 1H NMR.
Response: We have now provided the 1H chemical shifts up to 2 decimals throughout the manuscript.
- Check the resolution of the IR spectrometer. I think that it is 0.4 cm-1, and so quoting the frequency to 2 decimal places is not justified.
Response: We thank the reviewer for the careful evaluation. We have now corrected it in the revised manuscript.
- Descriptions of the treatment of the disorder in the crystal structures should be given in the experimental section.
Response: We have included the descriptions of the treatment of the disorder in the crystal structures in the revised experimental section.

Reviewer 3 Report
The presented manuscript “Trimetallic Polychalcogenide Species: Synthesis, Structures, and Aromaticity” is the next publications in the serious of articles of Sundargopal Ghosh & coauthors which deals with trimetallic clusters of niobium stabilized by tri/tetra-coordinated boron and mono/di-chalcogen moieties.
The two possible halcogenes were used in experiments selenium and sulfur. Structures of isolated molecules were confirmed by SCXRD, NMR and mass-spectroscopy.
All the described complexes contain an equilateral Nb3 triangular core stabilized by sulfur or selenium bridges. In three cases (complexes 1, 4 and II) additional boron-hydride bridge were presented.
It was shown that triniobium skeleton have 2-electron aromatic nature.
The manuscript is well written and the results obtained are reliable and do not cause any complaints.
Only two points I think can be improved.
The ratio of signals of two Cp* fragments in 2+3 and 4+5 NMR spectra authors take as “4:2”. Why not a “2:1”?
In the "References" chapter of manuscript present 99 point. And 43 of its are citations of S. Ghosh. I think it is too mach.
Author Response
The presented manuscript “Trimetallic Polychalcogenide Species: Synthesis, Structures, and Aromaticity” is the next publications in the serious of articles of Sundargopal Ghosh & coauthors which deals with trimetallic clusters of niobium stabilized by tri/tetra-coordinated boron and mono/di-chalcogen moieties. The two possible halogens were used in experiments selenium and sulfur. Structures of isolated molecules were confirmed by SCXRD, NMR and mass-spectroscopy. All the described complexes contain an equilateral Nb3 triangular core stabilized by sulfur or selenium bridges. In three cases (complexes 1, 4 and II) additional boron-hydride bridge were presented. It was shown that triniobium skeleton have 2-electron aromatic nature. The manuscript is well written and the results obtained are reliable and do not cause any complaints.
Response: We appreciate the reviewer's positive feedback and are happy to comment on the reviewer’s remarks.
Only two points I think can be improved.
- The ratio of signals of two Cp* fragments in 2+3 and 4+5 NMR spectra authors take as “4:2”. Why not a “2:1”?
Response: We thank the reviewer for the careful evaluation. We have corrected it in the revised manuscript.
- In the “References” chapter of manuscript present 99 point. And 43 of its are citations of S. Ghosh. I think it is too much.
Response: We have now revised the reference section in the revised manuscript and removed some of the references.
Round 2
Reviewer 2 Report
Although the authors have revised the manuscript, their responses to the referees' comments are not, on the whole satisfactory.
I have grave doubts about the crystal structure determination of 2. The authors solve the structure in the non-centrosymmetric monoclinic space group Cc, but the value of beta strongly suggests an orthorhombic crystal system and an apparent mirror plane bisecting both the organometallic molecule and the chloroform molecule. And rather than 58% and 42% occupancy of the sites, there should be 50% occupancy of each. In any case Rint is too high to give any confidence in the solution, and this structure cannot (should not) be published in its current state. And besides the obvious problems, the authors quote bond distances and angles to a high apparent accuracy - distances are given to four decimal places and one angle is given to three decimal places!
Scant details of the problems of the crystal structure determinations are given in either the main manuscript or the experimental details. There is no mention of twinning for 2, no mention of the solvent molecules, which one has to guess from the formulae. The formulae should be for individual compounds, not the total formula of the asymmetric unit.
Author Response
Response to reviewer
Reviewer: Although the authors have revised the manuscript, their responses to the referees' comments are not, on the whole satisfactory. I have grave doubts about the crystal structure determination of 2. The authors solve the structure in the non-centrosymmetric monoclinic space group Cc, but the value of beta strongly suggests an orthorhombic crystal system and an apparent mirror plane bisecting both the organometallic molecule and the chloroform molecule. And rather than 58% and 42% occupancy of the sites, there should be 50% occupancy of each. In any case Rint is too high to give any confidence in the solution, and this structure cannot (should not) be published in its current state. And besides the obvious problems, the authors quote bond distances and angles to a high apparent accuracy - distances are given to four decimal places and one angle is given to three decimal places!
Response: We thank the reviewer for his careful evaluation. The structure of compound 2 has been revised and is now solved and refined in space group Cmc21 as per the reviewer’s suggestion. Accordingly, we have corrected the bond distances and angles in the revised manuscript.
Reviewer: Scant details of the problems of the crystal structure determinations are given in either the main manuscript or the experimental details. There is no mention of twinning for 2, no mention of the solvent molecules, which one has to guess from the formulae. The formulae should be for individual compounds, not the total formula of the asymmetric unit.
Response: We thank again the reviewer for his careful evaluation. Now, we have provided all the details in the revised experimental section.